# Gulls in Porto Coastline as Reservoirs for *Salmonella* spp.: Findings from 2008 and 2023

**DOI:** 10.3390/microorganisms12010059

**Published:** 2023-12-28

**Authors:** Inês C. Rodrigues, Ana Paula Cristal, Marisa Ribeiro-Almeida, Leonor Silveira, Joana C. Prata, Roméo Simões, Paulo Vaz-Pires, Ângela Pista, Paulo Martins da Costa

**Affiliations:** 1ICBAS-UP—School of Medicine and Biomedical Sciences, University of Porto, Rua de Jorge Viterbo Ferreira, 228, 4050-313 Porto, Portugal; icrodrigues@icbas.up.pt (I.C.R.); annapcristal@hotmail.com (A.P.C.); up201008465@edu.icbas.up.pt (M.R.-A.); joana.prata@iucs.cespu.pt (J.C.P.); romeusimoes10@gmail.com (R.S.); vazpires@icbas.up.pt (P.V.-P.); 2CIIMAR—Interdisciplinary Centre of Marine and Environmental Research, Terminal de Cruzeiros do Porto, de Leixões, Av. General Norton de Matos s/n, 4450-208 Matosinhos, Portugal; 3UCIBIO—Applied Molecular Biosciences Unit, Laboratory of Microbiology, Department of Biological Sciences, Faculty of Pharmacy, University of Porto, 4050-313 Porto, Portugal; 4INSA—National Institute of Health, Department of Infectious Diseases, Av. Padre Cruz, 1649-016 Lisbon, Portugal; leonor.silveira@insa.min-saude.pt (L.S.); angela.pista@insa.min-saude.pt (Â.P.); 51H-TOXRUN—One Health Toxicology Research Unit, University Institute of Health Sciences, CESPU, CRL, 4585-116 Gandra, Portugal

**Keywords:** gulls, seagulls, *Salmonella* spp., *Laurus* spp., antimicrobial resistance, multidrug-resistant bacteria, whole-genome sequencing, serotyping

## Abstract

Gulls act as intermediaries in the exchange of microorganisms between the environment and human settlements, including *Salmonella* spp. This study assessed the antimicrobial resistance and molecular profiles of *Salmonella* spp. isolates obtained from fecal samples of gulls in the city of Porto, Portugal, in 2008 and 2023 and from water samples in 2023. Antimicrobial susceptibility profiling revealed an improvement in the prevalence (71% to 17%) and antimicrobial resistance between the two collection dates. Two isolate collections from both 2008 and 2023 underwent serotyping and whole-genome sequencing, revealing genotypic changes, including an increased frequency in the monophasic variant of *S.* Typhimurium. *qacE* was identified in 2008 and 2023 in both water and fecal samples, with most isolates exhibiting an MDR profile. The most frequently observed plasmid types were IncF in 2008 (23%), while IncQ1 predominated in 2023 (43%). Findings suggest that *Salmonella* spp. circulate between humans, animals, and the environment. However, the genetic heterogeneity among the isolates from the gulls’ feces and the surface water may indicate a complex ecological and evolutionary dynamic shaped by changing conditions. The observed improvements are likely due to measures to reduce biological contamination and antimicrobial resistance. Nevertheless, additional strategies must be implemented to reduce the public health risk modeled by the dissemination of pathogens by gulls.

## 1. Introduction

The association between humans, animals (companion animals, livestock, and wildlife), and the shared ecosystem facilitates the emergence and spread of zoonotic diseases [1]. Among the zoonotic pathogens, *Salmonella* spp. represent a public health threat due to their wide distribution, adaptability to infect both humans and animals, and the potential for carrying antibiotic resistance traits [2,3]. Salmonellosis is the second most reported foodborne disease in Europe [4]. The European Union implemented control programs in 2004 to target *Salmonella* serotypes based on the detection and management of infections in poultry production, processing, and distribution [5]. As a result, there was a notable decline in reported human cases between 2004 and 2021, namely from 200,000 to 60,050 cases [4,5,6]. Nevertheless, the environment still poses a risk for *Salmonella* spp. contamination because they can survive in various sources for extended periods of time, including in water from effluent discharge [1].

Antimicrobial resistance (AMR) carries severe implications for public health by reducing treatment efficacy, increasing mortality rates, and prolonging hospital stays [7,8]. It also has a significant impact on the environment, which is widely recognized as a reservoir for AMR, serving as a conduit for the dissemination of pre-existing resistant bacteria introduced through sewage, hospital wastewater, and agricultural waste and as a source and facilitator of their evolution [9,10]. The concern also extends to animals, where AMR affects the effectiveness of veterinary treatments but also contributes to the environmental reservoir, potentially facilitating the transmission of AMR bacteria and genetic elements to humans [11]. Therefore, there is an ongoing global health crisis driven by infectious diseases and the escalating challenge of AMR, which may be disseminated in the environment.

Marine wild birds like gulls are important carriers of *Salmonella* spp. due to their scavenging feeding behaviors and wide-ranging presence in different environments, and as such, they can be considered sentinel species [12]. In addition, gulls have been suggested as potential independent reservoirs capable of perpetuating and dispersing pathogenic and resistant strains [13,14]. The remarkable increase in gull populations, coupled with their ubiquity in urban and rural environments and their migratory capabilities, accelerates the dissemination of *Salmonella* spp. across diverse geographical regions and among a wide range of species, encompassing both humans and animals [15,16,17,18,19,20]. Indeed, recent studies conducted in northern Europe revealed an occurrence of *Salmonella* spp. in gulls of approximately 21%, and 19.2% of the *Salmonella* isolates exhibited multidrug-resistant (MDR) profiles [21]. Typhimurium was the most frequently observed serotype [21]. In contrast, Italy reported a prevalence of 1.3% of *Salmonella* spp. in gulls, and all strains were identified as *Salmonella arizonae*, with a predominant resistance to sulfonamides [13].

The objective of this study was to assess the antimicrobial resistance and molecular profiles of *Salmonella* spp. isolates obtained from fecal samples of gulls in the city of Porto, Portugal, in 2008 and 2023. Additionally, surface water samples were also collected from the same location in 2023, to evaluate environmental contamination. This study, to the best of our knowledge, is the first to evaluate the presence of *Salmonella* spp. in gull populations in the city of Porto.

## 2. Materials and Methods

### 2.1. Sample Collection

Fecal samples from wild gulls (*Laurus* spp.) were collected every 2 weeks during two distinct time sampling periods: December 2007 to April 2008 (hereinafter 2008) and December 2022 to April 2023 (hereinafter 2023). The sampling was performed in Porto, the second largest city of Portugal. The metropolitan region of Porto has a population of 1.7 million and a GDP of EUR 34.5 billion. Two sample points were selected (Figure 1): (i) Matosinhos beach (41°10′35.0″ N, 8°41′33.7″ W), a sandy beach bathed by Atlantic waters (predominantly NW currents) and north of the Douro River estuary; (ii) Largo António Calém (41°8’53.6″ N, 8°39’12.2″ W) in the northern margin of the Douro River estuary. These points were selected based on their proximity to a densely populated urban area, ease of access, and the frequent presence of gulls. Pools of 30 gull feces were collected in sterile tubes using a sterile spatula, careful to avoid sediment contamination. Water samples were also collected in 2023 from the same sampling points using 1 L sterile glass bottles to evaluate environmental contamination. Both the water and fecal samples were promptly placed in sterile containers and processed in the microbiology laboratory within 1 h. A total of 72 samples were analyzed: 24 fecal samples from 2008, 24 fecal samples from 2023, and 24 water samples from 2023.

### 2.2. Isolation of Salmonella spp.

Isolation of *Salmonella* spp. on both fecal and water samples was performed using conventional microbiological protocol recommended by ISO 6579 [22]. Briefly, fecal samples were precultured in buffered peptone water (BPW, Liofilchem, Teramo, Italy) at a 1/10 (*v*/*v*) dilution and incubated at 37 °C for 16 to 20 h. Next, 1 mL of the suspension was transferred into Mueller Kauffmann tetrathionate broth (MKT, Biokar Diagnostics, Allonne, France) and selenite cystine broth (SC, Merck, Darmstadt, Germany). Samples from the MKT and SC were inoculated on selective agars: Hektoen enteric agar (HEA, Biokar Diagnostics) and xylose lysine deoxycholate agar (XLD, Biokar Diagnostics) and incubated at 37 °C for 24 h.

Regarding water samples, a volume of 40 mL was filtered through 0.45 µm pore size membrane filters (Whatman, Maidstone, UK). The filtered membrane was placed in 10 mL of BPW (Liofilchem, Waltham, MA, USA) at 37 °C for 24 h. After the incubation, 1 mL of the suspension was added to MKT and SC, and 10 µL were inoculated in MRSV. Then, the samples were inoculated on both HEA and XLD. After selective isolation on HEA and XLD, bacterial isolates from both fecal and water samples underwent biochemical tests for lactose fermentation and urea hydrolysis using triple sugar iron agar (TSI, Biokar Diagnostics, Allonne, France) and motility indole urea (MIU, Liofilchem) agars, respectively, to differentiate *Salmonella* spp. from other non-lactose-fermenters of the family *Enterobacteriaceae*. Bacterial isolates were further subjected to a rapid latex agglutinations test (Thermo Scientific™ Oxoid™ *Salmonella* Test Kit, Waltham, MA, USA). Pure colonies of *Salmonella* spp. from both fecal and water samples were stored in BPW supplemented with 1.5% glycerol (Biokar Diagnostics) at −20 °C for further analysis.

### 2.3. Salmonella Identification by Polymerase Chain Reaction (PCR)

Prior to polymerase chain reaction (PCR), DNA extraction from fresh and pure colonies was performed by suspending a colony in 20 μL of TE buffer (Tris 10 mM + EDTA 1 mM, pH = 8) and incubating for 15 min at 95 °C using a dry block heating thermostat (BIO TDB-100, Biosan, Riga, Latvia). Afterwards, a volume of 180 μL of sterile ultrapure water was added to the suspension and centrifuged at 12,000 rpm for 3 min. The supernatant was stored at −20 °C. 

PCR was chosen over other molecular techniques due to its specificity in *Salmonella* spp. identification by targeting the *invA* gene [23]. The detection of the *invA* gene was performed using the primers InvA_R (5′GTGAAATTATCGCCACGTTCGGGCAA) and InvA_F (5′TCATCGCACCGTCAAAGGAACC) [23]. The PCR was performed in a 25 μL reaction mixture containing 12.5 μL of Master Mix (2× DreamTaq Hot Start PCR Master Mix, NZYTech, Lisboa, Portugal), 1 μL each of forward and reverse primers (10 μM), and 4 μL of bacterial DNA. In a thermal cycler (MyCycler™, Biorad, Hercules, CA, USA), the PCR conditions were as follows: an initial denaturation at 95 °C for 3 min, followed by 35 cycles of denaturation at 94 °C for 30 s, 54 °C for 30 s, 72 °C for 1 min, and a final extension at 72 °C for 7 min. Genomic DNA of *Salmonella typhimurium* CECT 443 was used as control in the PCR assay.

The 284-bp fragments present were subjected to electrophoresis on 1.5% (*w*/*v*) agarose gel (Agarose Ultrapure grade, NZYTech) in 1×TBE at 100V for 45 min and stained with Green Safe Premium (NZYTech). Ladder VII (NZYTech) was used as the molecular weight marker.

### 2.4. Antimicrobial Susceptibility Testing

The resistance patterns of all *Salmonella* spp. isolates were determined with the Kirby–Bauer method on Mueller–Hinton agar (MHA, Biokar Diagnostics) following the Clinical Laboratory Standards guidelines [24]. The following antimicrobial drugs were used: amoxicillin–clavulanate (AMC, 30 µg), amikacin (AMK, 30 µg), ampicillin (AMP, 10 µg), aztreonam (ATM, 30 µg), cefazoline (CFZ, 30 µg), cefoxitin (FOX, 30 µg), cefotaxime (CTX, 30 µg), ceftazidime (CAZ, 30 µg), chloramphenicol (CHL, 30 µg), ciprofloxacin (CIP, 5 µg), doxycycline (DOX, 30 µg), gentamycin (GEN, 10 µg), imipenem (IPM, 10 µg), levofloxacin (LEV, 5 µg), nitrofurantoin (NIT, 30 µg), streptomycin (STR, 10 µg), sulfamethoxazole–trimethoprim (SXT, 25 µg), tetracycline (TET, 30 µg), and tobramycin (TOB, 10 µg). All antimicrobial disks were from Oxoid (Basingstoke, UK). *Escherichia coli* ATCC 25922 was used as reference strain. All bacterial isolates were classified as susceptible, intermediate, or resistant, using the current CLSI breakpoints [24]. Isolates resistant to 3 or more antibiotics classes were defined as multidrug-resistant (MDR) bacteria [25].

In addition, the colistin resistance was assessed by determining the minimal inhibitory concentrations (MICs) through the broth microdilution method as recommended by EUCAST [26]. Briefly, fresh colonies were suspended in cation-adjusted Mueller–Hinton broth (CAMHB, Sigma-Aldrich, St Louis, MO, USA), and optimal density at 600 nm was adjusted to 0.1. A final inoculum concentration of 5 × 10^5^ colony-forming unit per mL (CFU/mL) was achieved in each well containing two-fold serial dilutions of colistin (concentrations ranged from 1 to 16 µg/mL) in a sterile 96-well U-shaped untreated polystyrene plate. Microplates were incubated for 16–20 h at 37 °C and the MIC was determined as the lowest concentration of colistin that prevented visible growth. Positive (without colistin) and negative (without inoculum) controls were used. At least three independent assays were conducted.

Statistical analysis was conducted on IBM SPSS Statistics 29, namely, using Fisher’s exact test (or its extension for tables larger than 2 × 2, the Fisher–Freeman–Halton test) to compare the frequencies of resistance strains between samples, considering α = 0.05. This test was preferred to Chi-square due to the presence of low expected values (<5) in several cells.

### 2.5. Creating a Salmonella Collection of 2008 and 2023

At least one *Salmonella* spp. isolate from each positive sample in 2008 and 2023 was selected for serotyping and whole-genome sequencing (WGS) according to the antimicrobial susceptibility profiles.

#### 2.5.1. Serotyping of *Salmonella* Isolates

The selected *Salmonella* isolates were serotyped using the slide agglutination method for somatic and flagellar antigens (SSI Diagnostica, Hillerod, Denmark; Sifin diagnostics, Berlin, Germany), according to the Kauffmann–White–Le Minor scheme [27].

#### 2.5.2. WGS Characterization and Bioinformatics Analysis

Genomic DNA extraction was performed using the ISOLATE II genomic DNA kit (Bioline, London, UK) and quantified in the Qubit fluorometer (Invitrogen, Waltham, MA, USA) with the dsDNA HS assay kit (Thermo Fisher Scientific, Waltham, MA, USA), following the manufacturer’s guidelines. The NexteraXT library preparation protocol (Illumina, San Diego, CA, USA) was performed prior to cluster generation and paired-end sequencing (2 × 150 bp or 1 × 250 bq) on a MiSeq or a NextSeq 550 instrument (Illumina) according to the manufacturer’s instructions. Sequencing reads were submitted to the QAssembly pipeline (v 3.61) of EnteroBase (https://enterobase.warwick.ac.uk/; accessed on 1 July 2023) for quality control, trimming, and to generate assemblies of high quality.

Online bioinformatic tools from EnteroBase v5.1 (https://enterobase.warwick.ac.uk//; accessed on 3 July 2023), were used to determine the sequence type (ST) and the core genome multilocus sequence typing (cgMLST). Tools from the Center for Genomic Epidemiology (CGE, http://www.genomicepidemiology.org; accessed on 31 July 2023) were also used to assess antibiotic resistance genes and point mutations (ResFinder 4.1); and plasmid replicons (PlasmidFinder 2.1). Virulence genes were obtained from the Virulence Factors of Pathogenic Bacteria (VFDB) platform (http://www.mgc.ac.cn/cgi-bin/VFs/genus.cgi?Genus=Salmonella; accessed on 1 July 2023), using the Vfanalyser tool. A comparative genomic analysis was made using the core genome MLST (cgMLST) of the isolates in this study and of additional genomes obtained from EnteroBase (https://enterobase.warwick.ac.uk/; accessed on 1 July 2023), according to the country (Portugal) and the year (2008 and 2023). The cgMLST analysis was based on the *Salmonella* schemes provided by EnteroBase, encompassing 3002 loci [28] (https://enterobase.warwick.ac.uk/; accessed on 10 July 2023) and the hierarchical clustering of cgMLST (HierCC) [29]. Based on these data, a minimum spanning tree was constructed using both GrapeTree and NINJA NJ tools [30].

#### 2.5.3. Data Availability

The sequence reads were submitted to the European Nucleotide Archive (ENA) under BioProject accession number PRJEB32515. The accession numbers of the genomic sequences for each strain are listed in the Appendix A.

## 3. Results

### 3.1. Detection of Salmonella spp.

In 2008, a total of 77 isolates were obtained from 17 (71%) fecal samples, while in 2023, 11 isolates were identified from 4 (17%) fecal samples and 7 were recovered from 3 (13%) water samples (Table 1).

### 3.2. Antimicrobial Susceptibility of Salmonella spp. Isolates

The antimicrobial susceptibility results of the *Salmonella* spp. isolates are displayed in Table 2 and Figure 2. While categorized into resistant, intermediate, and susceptible groups, the intermediate susceptibility results are presented and discussed as resistant, simplifying the assessment of the relative frequency of antimicrobial resistance.

Regarding *Salmonella* spp. isolates recovered from fecal samples collected in 2008, the predominant resistances were mainly toward streptomycin (70%), tetracycline and doxycycline (39%), and ampicillin (27%). Among the 2023 collection, the leading resistances were against streptomycin (83%), tetracycline and doxycycline (44%), ampicillin (44%), and sulfamethoxazole–trimethoprim (11%). No antimicrobial resistance to colistin, fluoroquinolones, monobactams, nitrofurans, or carbapenems were observed in either collection (Appendix B, Table A1).

When comparing the relative percentages of antibiotic resistance in isolates from the fecal samples from both collection dates, resistance to cefazolin, chloramphenicol, gentamicin, sulfamethoxazole–trimethoprim, and tobramycin was not observed in 2023, and tetracycline resistance showed a small decrease (36% versus 39%). Furthermore, the relative percentage of resistance to ampicillin (from 27% to 55%), doxycycline (from 27% to 55%), and streptomycin (from 27% to 55%) was higher in 2023. However, no statistically significant differences were observed (Appendix A). Resistance to tetracycline was an exception, presenting a significant difference between the two years (*p* = 0.018) due to a relative increase in the intermediate resistant isolates and a decrease in the resistant and susceptible isolates in 2023. However, statistical analysis was limited by the lower number of isolates from 2023 stemming from the decrease in *Salmonella* prevalence.

Comparing water samples and fecal samples from 2023, sulfamethoxazole–trimethoprim presented a 29% resistance rate in the water samples, while the fecal isolates displayed complete susceptibility. There was no statistically significant difference in antibiotic resistance between the fecal and the water samples (Appendix A).

MDR profiles were observed in 18 isolates (23%) from 2008. In 2023, MDRs were detected in 8 (44%) from 2023, of which 6 (55%) were from fecal samples and 2 (29%) from water.

### 3.3. Molecular Analysis

#### 3.3.1. Creating Salmonella Collections

A collection of *Salmonella* spp. isolates was created for each sampling time (2008 and 2023) based on the following criteria: (i) inclusion of at least one *Salmonella* spp. isolate per sample and (ii) inclusion of all isolates with distinct antimicrobial susceptibility profiles within the same sampling time. Therefore, 26 isolates from 2008 and 7 from 2023 were selected for serotyping and WGS. Antimicrobial resistance determinants, ST, cgMLST, and plasmid replicons of *Salmonella enterica* from the 2008 and 2023 collections are displayed in Table 3 and Table 4, respectively. Virulence factors are detailed in the Appendix A.

#### 3.3.2. *Salmonella* Serotyping

A total of 13 distinct serotypes of *Salmonella enterica* were identified in 2008. The *S.* Typhimurium serotype was the predominant serotype (35%, 9/26), followed by *S.* Derby (15%, 4/26), *S.* Enteritidis (12%, 3/26), and a monophasic variant of *S.* Typhimurium 1,4,[5],12:i:- (8%, 2/26) (Table 3). Among the 2023 isolates, only six different serotypes were observed, with monophasic variants of *S.* Typhimurium 1,4,[5],12:i:- and 1,4,[5],12:b:- being the most prevalent (43%, 3/7) (Table 4). When comparing the 2023 selected isolates from the water and fecal samples, the monophasic variant of *S.* Typhimurium was more frequently detected in fecal samples (75%, 3/4) but not in water. Surface water samples exhibited different serotypes (*S.* Saintpaul, *S*. Rissen, and *S*. Poona), which were not identified in the fecal samples.

#### 3.3.3. WGS Characterization

##### Sequence Type Determination

In the 2008 collection, the most frequent STs were ST19 (27%, 7/26), ST34 (15%, 4/26), ST40 (15%, 4/26), and ST11 (8%, 3/26) (Table 3). Among the 2023 samples, ST34 was the most frequently detected in fecal isolates (50%, 2/4), while no reoccurring ST was identified in isolates from water (Table 4).

##### Antimicrobial Resistance Determinants

All isolates from the 2008 collection displayed at least one acquired resistance gene commonly associated with aminoglycoside resistance (*aac(2′)-Iia*, *aac(6′)-Iaa*, *aph(6)-Id*, *aph(3″)-Ib*, *aph(4)-Ia*, *aadA1*, *aac(3)-IV*, and *aadA2b*) (Table 3). Eleven isolates showed resistant genes associated with tetracyclines (*tet(A)*, *tet(B)*) and sulfonamides (*sul1*, *sul2*), while nine isolates displayed predictive resistance to β-lactams (*bla*_TEM-1B_*).* Seven isolates harbored acquired genes linked to resistance to trimethoprim (*drfA14*, *dfrA1*), and five isolates carried genes associated with fosfomycin (*fosA7*) resistance. Additionally, two isolates displayed an antiseptic-resistant gene (*qacE*) and one demonstrated a florfenicol-related resistance gene (*floR*).

Similar to the 2008 collection, all isolates from 2023 presented at least one acquired gene related to aminoglycoside resistance (*aac(6′)-Iaa*, *aph(6)-Id*, *aph(3″)-Ib*, *aadA1,* and *aadA2)* (Table 4). Additionally, within the 2023 collection, three isolates carried acquired genes related to penicillins (*bla*_TEM-1B_*)* and sulfonamides (*sul1*, *sul2*) resistance, while another set of two isolates harbored predictive trimethoprim resistance (*drfA12*). Acquired genes conferring predictive resistance to macrolides (*mph(A)*) was found in two other isolates.

No genes or mutations mediating resistance to carbapenems, colistin, monobactams, or nitrofurans were found in either the 2008 or 2023 collections. Genes conferring resistance to fosfomycin (*fosA7)* and phenicols (*floR*) were detected only in the 2008 isolates collection.

Furthermore, the phenotypic and genotypic results were consistent (Appendix A).

##### Virulence Factors

According to VFDB, 216 genes from 14 different virulence factor classes were identified, with a minimum of 74 and a maximum of 163 genes found in an isolate (Appendix A). Fifty-four genes from six VF classes were found in all genomes. As expected, genes associated with the Vi antigens were not found in any isolate. The gene-encoding exotoxin SpvB was identified in five isolates from 2008 (three *S*. Enteritidis and two *S*. Typhimurium), while the *cdtB* and *pltA* genes, belonging to the typhoid toxin group, were both present in five isolates, four from 2008 (*S*. Brandenburg, *S*. Panama, *S*. Give, and *S*. Bredeney) and one from 2023 (*S*. Poona). Other relevant genes, *invA*, *hliA,* and *fimA,* were identified in 100%, 97%, and 61% of isolates, respectively.

##### Plasmid Replicons

WGS allowed the identification of 11 different plasmids in both the 2008 and 2023 isolates (Table 3 and Table 4). IncF was the most prevalent plasmid replicon in 2008 (23%, 6/26), while IncQ1 was the most frequent in 2023 (43%, 3/7).

##### Phylogenetic Analysis

The comparative analysis of gull isolates using hierarchical clustering with a cut-off HierCC score ≤ 5 detected three distinct clusters among the isolates from 2008 (Figure 3, red circles), each presenting a high degree of phylogenetic relatedness. Two *Salmonella* Derby (PT_SE0336 and PT_SE0173) share a common HC5 cluster (HC5|336752) (α-circle), with three allelic differences between them. Furthermore, two other *S*. Typhimurium clusters were formed (β-circles): β1 (HC2|336817), composed of three clones (PT_SE0245, PT_SE0340, PT_SE0342) with one allelic difference; and β2 (HC2|336818), composed of two clones (PT_SE0243, PT_SE0244) with two allelic differences.

Phylogenetic analysis (Appendix A) does not reveal any close phylogenetic relation between gull and human isolates or gull and surface water 2023 isolates.

## 4. Discussion

This study aimed to assess the antimicrobial resistance and molecular profiles of *Salmonella* spp. isolates in gull fecal samples from Porto, Portugal in 2008 and 2023, while also investigating surface water samples from 2023. Previous research has indicated an increased risk of transmitting pathogens, such as *Salmonella* spp., *Campylobacter* spp., and *Chlamydia* spp., between infected gulls and human populations [31]. *Salmonella* spp. was detected in both sampling years, but the positivity was lower in 2023 (17%) than in 2008 (71%). Nevertheless, the prevalence found in 2023 in Porto (17%) is higher than that found in previous years (2020–2021) in three other locations in Portugal (2.8%) [20] and in southern Italy (1.3%; 2016 to 2019) [13]. The prevalence in surface water samples from 2023 was 13%. In 2013, *Salmonella* spp. have been previously isolated in two lakes in Porto, presenting high susceptibility to all antibiotics tested [32].

Antimicrobial resistance was generally lower in 2023. Gulls, as sentinels for antimicrobial resistance and carriers of *Salmonella* spp., may offer valuable insights into the environmental impact of measures addressing AMR in human and veterinary medicine [33,34,35]. Regulations aiming to control this serious public health problem have been in place since 2006, when the European Union prohibited the use of antibiotics as growth promoters [36]. Additional measures were taken in 2020, with the implementation of a categorization system for antibiotics to restrict the use of those critically important for human medicine [37]. For example, streptomycin and ampicillin were categorized as critically important antibiotics to human medicine. The categorization of both antibiotics was based on the need for their use in treatment of infections by Enterobacteriaceae from nonhuman sources and on the availability of options to treat serious infections, for example, the use of streptomycin on enterococcal endocarditis, MDR tuberculosis, and MDR Enterobacteriaceae and of ampicillin on *Listeria* and *Enterococcus* spp. infections [38]. In Portugal, community use of antibiotics in medicine fell from 18.7 to 13.7 defined daily doses per 1000 inhabitants per day between 2012 and 2021 [39], while veterinary use decreased from 166 to 150 mg per population corrective unit between 2010 and 2021 [40] (Appendix B, Figure A1 and Figure A2).

Resistance to antibiotics from category C (caution) that was observed in gull feces isolates from 2008 were no longer found in 2023, except for streptomycin, which increased (29% to 55%). Moreover, 29% of the isolates from water were resistant to sulfamethoxazole–trimethoprim, while all isolates presented resistance or intermediate resistance to streptomycin, suggesting that water may be a potential reservoir of antimicrobial resistance genes. An increase in resistance to antibiotics from category D (precaution) was observed between the two collections, namely, to doxycycline (39% to 55%) and ampicillin (27% to 55%). These results are in accordance with an increased use of aminoglycosides and penicillin in veterinary medicine in Portugal, but not with a decreased use of tetracycline [40]. The decreasing trend in resistance to antibiotics from categories B and C and the increasing trend in resistance to category D antibiotics is in accordance with those documented on livestock in European member states [41]. Despite European regulations on antibiotic use, MDR isolates were identified in both years, exhibiting a higher relative percentage in 2023 (23% and 44% in 2008 and 2023, respectively) but lower absolute frequency (18/77 against 8/18 in 2023). This suggests an increased likelihood of encountering MDR strains despite an overall improving trend in antimicrobial resistance. In 2023, surface water and gull feces presented *Salmonella* spp. with resistance to two critically important antibiotics to human medicine, streptomycin and ampicillin.

Regarding the collections of both 2008 and 2023, the phenotypic antimicrobial sensitivity results and WGS analysis were in alignment. In addition, antiseptic-resistant genes (*qacE*) were also identified in 2008 and 2023 in both the water and fecal samples, most isolates exhibiting an MDR profile (75%). Indeed, bacterial exposure to quaternary ammonium compounds may result from household, industrial, and clinical uses (e.g., present in effluents or waste) [42], and it could lead to antibiotic resistance through the selection of class 1 integrons [43]. The most frequently observed plasmids (23%) of *Salmonella* isolates from 2008 were IncF (IncFIB and IncFII), which are widely distributed in the *Enterobacteriaceae* family [44] and may carry bacterial determinants of virulence [44,45]. Conversely, IncQ1 was predominant (43%) in 2023, which could be involved in the dissemination of tetracycline resistant genes [46].

There was a decrease in the diversity of *Salmonella* serotypes found in gulls’ feces in 2023 compared with 2008. The monophasic variant of *S.* Typhimurium was predominant in 2023 (50%) compared with the common serotypes identified in 2008, which included *S.* Typhimurium, *S.* Derby, and *S.* Enteritidis. These serovars have been associated with foodborne illnesses in the European Union, namely, *S.* Enteritidis (54.6%), *S.* Typhimurium (11.4%), the monophasic variant of *S.* Typhimurium (8.8%), and *S.* Derby (0.93%) [4]. Additionally, a cluster of the monophasic variant of *S.* Typhimurium was detected in clinical samples from Porto spanning the years 2001 to 2011 [47]. Despite the isolation of five isolates of a monophasic variant of *S.* Typhimurium from both collections, only one exhibited multidrug resistance (MDR). It is worth noting that this serotype often exhibits MDR profiles [48,49].

The predominant sequence type in 2008 was ST19, likely due to the frequency of the *S.* Typhimurium [50]. In 2023, ST34 was the most frequent type, linked to a monophasic variant of *S*. Typhimurium [50], which has been associated with foodborne outbreaks in Europe and China [51,52]. Moreover, *S.* Rissen ST469 was identified in samples from 2008 (PT_SE0344) and 2023 (PT_SE0354). While serovar Rissen ST469 is not frequently associated with clinical cases, it has been isolated previously in the Azores between 2014 and 2017, suggesting that this serovar is disseminated in the Portuguese islands and the mainland [53].

No hierarchical clustering (≤5 alleles) between feces and water was found, likely resulting from the difficulty in relating ecologically distinct populations of *Salmonella* spp., considering the impact of genetic drift, plasmid exchange, and geographical isolation on their genomes [54]. Nonetheless, the environment has already been proven to play an important role in the propagation of *Salmonella* spp. and antimicrobial resistance [55]. While no distinct pathway of environmental exposure has been identified, the presence of serotypes previously associated with human infections (e.g., *S.* Typhimurium) poses a public health threat [56]. Moreover, resistance strains were also isolated from the Douro River’s surface waters. Indeed, improvements in wastewater treatment infrastructure (e.g., an increase in tertiary treatment from 24% to 51% between 2009 and 2018) could have been undermined by emergency discharge of untreated effluents [57].

Gulls could be exposed to *Salmonella* through contact with effluents, waste, and other animal species [58]. The proximity to a high-population-density area favor scavenging feeding habits, which expose birds to urban waste and human food sources. Indeed, more than 81% of gull pellets in Porto contained anthropogenic debris (i.e., glass, plastic) in 2018 [59]. Migratory behaviors could further contribute to the dissemination of serotypes and resistance genes across borders (e.g., along the European and African Atlantic coasts) [60]. Thus, gulls might increase the risk of transmission to humans through the contamination of water, fishing ports or fishing farms, and beaches [31]. In fact, gulls (*Laurus* spp.) have previously been associated with the spread of AMR *Escherichia coli* from a landfill to a river [61].

The city of Porto is populated by 593 to 813 *Laurus michahellis* [62]. Gulls, as autochthone and migratory species, are protected by national law, which means that population control, through culling or the destruction of eggs or nests, is prohibited [63]. Therefore, population control relies solely on controlling food sources, using deterrents (e.g., placing bird spikes on building facades), and increasing awareness [64]. The potential for the dissemination of pathogens may challenge this stance, even though gulls can act as either a source or as a vector. A decrease in *Salmonella* spp. prevalence and antibiotic resistance suggests that combined measures (e.g., restrictions on antibiotic use, wastewater treatment) may reduce the role of gulls as reservoirs. Either way, gulls remain valuable indicators of microbial status, bridging the gap between human populations and the environment.

## 5. Conclusions

Successful interdisciplinary measures on reducing biological contamination and antimicrobial use contribute to the decrease in the prevalence of *Salmonella* spp. and antibiotic resistance. Yet, we show that surface waters of the Douro River, as well as gulls’ feces, harbor resistant strains of *Salmonella* serovars usually associated with human infections. An increased frequency of the monophasic variant of S. Typhimurium was also observed, raising public health concerns due to its association with multidrug-resistant (MDR) profiles and outbreaks throughout Europe and China. *Salmonella* spp. are evidenced to be circulating between the environment, animals, and human populations. However, source identification is still a challenge for several reasons, including the continuous genotypic and phenotypic changes in *Salmonella* spp. while adjusting to their environment. Limiting the exposure of gulls to biological contamination and implementing population control in areas of high population density, by reducing scavenging opportunities, for example, could help reduce the public health risk posed by these reservoirs and/or vectors of pathogenic agents.

## Figures and Tables

**Figure 1 microorganisms-12-00059-f001:**
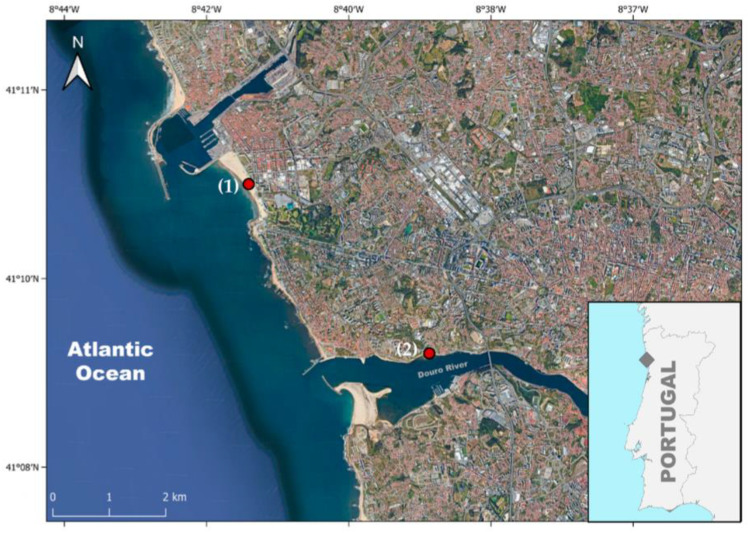
Map of the sample collection points, where (1) represents Matosinhos beach and (2) Largo António Calém, in Porto, Portugal. Map created using the Free and Open Source QGIS.

**Figure 2 microorganisms-12-00059-f002:**
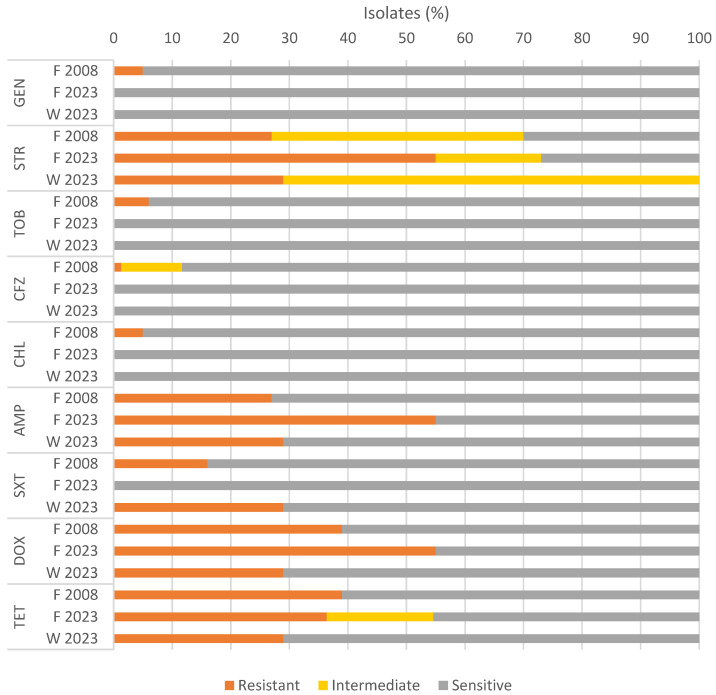
Antimicrobial susceptibility of isolates from gulls’ feces (F 2008 and F 2023) and water samples from the Douro River (W 2023). Aminoglycosides: gentamycin (GEN), streptomycin (STR), tobramycin (TOB). Cephalosporins: cefazoline (CFZ). Miscellaneous: chloramphenicol (CHL). Penicillin: ampicillin (AMP). Sulfonamides: sulfamethoxazole–trimethoprim (SXT). Tetracycline: tetracycline (TET), doxycycline (DOX).

**Figure 3 microorganisms-12-00059-f003:**
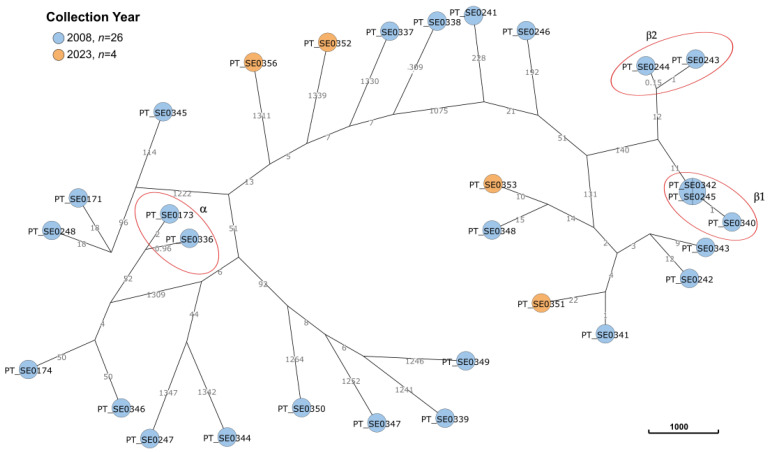
Comparative GrapeTree analysis of the 2008 and 2023 *Salmonella* isolates recovered from gulls’ feces. The core genome minimum spanning tree was created within the EnteroBase pipeline using the NINJA NJ algorithm and GrapeTree tool, comprising a total of 3002 target loci of *Salmonella*. The scale bar corresponds to the number of cgMLST allelic differences. Clusters are marked with red circles.

**Table 1 microorganisms-12-00059-t001:** Overview of the total samples collected, the percentage of positive samples, and the number of *Salmonella* spp. isolates recovered from 2008 and 2023 samplings.

		Sampling (*n*)	Number of Positive Samples	Number of *Salmonella* spp. Detected
2008	F	24	17 (71%)	77
2023	F	24	4 (17%)	11
	W	24	3 (13%)	7

**Table 2 microorganisms-12-00059-t002:** Summary of the percentage of phenotypic antibiotic resistance of *Salmonella* spp. isolates recovered from gull feces and surface water from Porto, Portugal, in 2008 and 2023.

Year	Samples	*n*	*Salmonella* spp. Isolates	AMP			CFZ			CHL			DOX			GEN			STR			SXT			TET			TOB		
				R	I	S	R	I	S	R	I	S	R	I	S	R	I	S	R	I	S	R	I	S	R	I	S	R	I	S
**2008**	F	24	77	27	0	73	1	10	88	5	0	95	39	0	61	5	0	95	27	43	30	16	0	84	39	0	61	7	0	94
**2023**	F	24	11	55	0	45	0	0	100	0	0	100	55	0	46	0	0	100	55	18	27	0	0	100	36	18	46	0	0	100
	W	24	7	29	0	71	0	0	100	0	0	100	29	0	71	0	0	100	29	71	0	29	0	71	29	0	71	0	0	100

F: feces; W: water; n: number of samples; AMP: ampicillin; CFZ: cefazoline; CHL: chloramphenicol; DOX: doxycycline; GEN: gentamycin; STR: streptomycin; SXT: sulfamethoxazole–trimethoprim; TET: tetracycline; TOB: tobramycin; R: resistant; I: intermediate; S: susceptible.

**Table 3 microorganisms-12-00059-t003:** Serotype, ST, cgMLST, phenotypic antimicrobial resistance profile, antimicrobial resistance determinants, and plasmid replicons of the selected *Salmonella* isolates from gull feces in 2008.

Isolate	Serotyping	ST	cgMLST	Phenotypic Antimicrobial Resistance Profile	Resistance Genes	Plasmid
PT_SE0338	Braenderup	22	378210	None	*aac(6′)-Iaa*, *aac(2′)-Iia*, *parC*:T57S	NF
PT_SE0339	Brandenburg	10,807	378218	STR^I^	*aac(6′)-Iaa*, *parC*:T57S	NF
PT_SE0350	Bredeney	897	378215	CHL^R^, DOX^R^, GEN^R^, STR^I^, SXT^R^, TET^R^, TOB^R^ *	*aac(6′)-Iaa*, *aph(3″)-Ib*, *aph(4)-Ia*, *aph(6)-Id*, *aadA1*, *aac(3)-IV*, *floR*, *dfrA1*, *parC:*T57S, *qacE*, *sul1*, *tet(A)*	ColpVC, IncHI1A(NDM-CIT)
PT_SE0336	Derby	40	378219	DOX^R^, STR^I^, TET^R^	*aac(6′)-Iaa*, *fosA7*, *tet(B)*, *parC*:T57S	NF
PT_SE0173	Derby	40	336752	DOX^R^, STR^I^, TET^R^	*aac(6′)-Iaa*, *fosA7*, *tet(B)*, *parC*:T57S	NF
PT_SE0174	Derby	40	336759	STR^I^	*aac(6′)-laa*, *fosA7*, *parC*:T57S	NF
PT_SE0346	Derby	40	378211	DOX^R^, STR^R^, TET^R^	*aac(6′)-Iaa*, *aadA2b*, *fosA7*, *parC*:T57S, *qacE*, *sul1*, *tet(A)*	Col(pHAD28)
PT_SE0248	Enteritidis	11	336819	None	*aac(6′)-Iaa*	IncFIB, IncFII
PT_SE0345	Enteritidis	11	378224	None	*aac(6′)-Iaa*	IncFIB(S), IncFII(S)
PT_SE0171	Enteritidis	11	336751	None	*aac(6′)-Iaa*	IncFIB, IncFII
PT_SE0347	Give	7704	378214	None	*aac(6′)-Iaa*, *parC*:T57S	NF
PT_SE0337	London	155	378209	None	*aac(6′)-Iaa*, *parC*:T57S	NF
PT_SE0343	1,4,[5],12:i:-	34	378212	AMP^R^, STR^R^	*aac(6′)-Iaa*, *aph(3″)-Ib*, *aph(6)-Id*, *bla*_TEM-1B_, *sul2*	IncQ1
PT_SE0348	1,4,[5],12:i:-	34	378220	DOX^R^, STR^R^, TET^R^	*aac(6′)-Iaa*, *aph(3″)-Ib*, *aph(6)-Id*, *bla*_TEM-1B_, *sul2*, *tet(B)*	Col156, IncFII(pRSB107), IncQ1
PT_SE0349	Panama	48	378216	None	*aac(6′)-Iaa*, *parC*:T57S	NF
PT_SE0344	Rissen	469	378221	DOX^R^, TET^R^	*aac(6′)-Iaa*, *tet(A)*, *parC*:T57S	NF
PT_SE0247	Tennessee	319	336860	None	*aac(6′)-Iaa*, *fosA7*, *parC*:T57S	NF
PT_SE0241	Typhimurium	19	336815	STR^I^	*aac(6′)-Iaa*	ColpVC, IncFIB, IncFII
PT_SE0243	Typhimurium	19	336818	AMP^R^, CFZ^I^, DOX^R^, STR^I^, SXT^R^, TET^R^ *	*aac(6′)-Iaa*, *aph(3″)-Ib*, *aph(6)-Id*, *bla*_TEM-1B_, *dfrA14*, *sul2*, *tet(A)*	IncN
PT_SE0244	Typhimurium	19	336820	AMP^R^, CFZ^I^, DOX^R^, STR^I^, SXT^R^, TET^R^ *	*aac(6′)-Iaa*, *aph(3″)-Ib*, *aph(6)-Id*, *bla*_TEM-1B_, *dfrA14*,, *sul2*, *tet(A)*	IncN
PT_SE0245	Typhimurium	19	336817	AMP^R^, CFZ^I^, DOX^R^, SXT^R^, TET^R^ *	*aac(6′)-Iaa*, *aph(3″)-Ib*, *aph(6)-Id*, *bla*_TEM-1B_, *dfrA14*, *sul2*, *tet(A)*	IncN
PT_SE0340	Typhimurium	19	378211	AMP^R^, DOX^R^, SXT^R^, TET^R^ *	*aac(6′)-Iaa*, *aph(3″)-Ib*, *aph(6)-Id*, *bla*_TEM-1B_, *dfrA14*, *sul2*, *tet(A)*	IncN
PT_SE0342	Typhimurium	19	378222	AMP^R^, CFZ^R^, DOX^R^, STR^I^, SXT^R^, TET^R^ *	*aac(6′)-Iaa*, *aph(3″)-Ib*, *aph(6)-Id*, *bla*_TEM-1B_, *dfrA14*, *sul2*, *tet(A)*	IncN
PT_SE0246	Typhimurium	19	336821	STR^I^	*aac(6′)-Iaa*	IncFIB, IncFII
PT_SE0242	Typhimurium	34	336816	AMP^R^, STR^R^	*aac(6′)-laa*, *aph(3″)-Ib*, *aph(6)-Id*, *bla*_TEM-1B_, *sul2*	IncQ1
PT_SE0341	Typhimurium	34	378223	AMP^R^, DOX^R^, STR^R^, TET^R^ *	*aac(6′)-Iaa*, *aph(3″)-Ib*, *aph(6)-Id*, *bla*_TEM-1B_, *dfrA14*, *sul2*, *tet(B)*	IncQ1

^R^ Resistant; ^I^ Intermediate; * MDR profile; AMC: amoxicillin/clavulanic acid; AMP: ampicillin; CFZ: cefazoline; CHL: chloramphenicol; DOX: doxycycline; NIT: nitrofurantoin; STR: streptomycin; SXT: sulfamethoxazole–trimethoprim; TET: tetracycline; TOB: tobramycin; NF: not found.

**Table 4 microorganisms-12-00059-t004:** Sample type, serotype determination (SISTR1 Serovar), ST, cgMLST, phenotypic antimicrobial resistance profile, antimicrobial resistance determinants (ResFinder), and plasmid replicons (PlasmidFinder) of the selected *Salmonella* isolates from gull feces and surface water from 2023.

Isolate	Sample Type	Serotyping	ST	cgMLST	Antimicrobial Resistance Profile	Resistance Genes	Plasmid
PT_SE0356	F	Bovismorbificans	142	366510	STR^I^	*aac(6′)-Iaa*	Col (pHAD28), Col156, Col440I_1
PT_SE0351	F	1,4,[5],12:i:-	34	366505	AMP^R^, STR^R^	*aac(6′)-Iaa*, *aph(6)-Id*, *aph(3″)-Ib*, *bla*_TEM-1B_, *sul2*	Col (pHAD28), IncQ1; p0111
PT_SE0353	F	1,4,[5],12:b:-	34	366511	AMP^R^, DOX^R^, STR^R^, TET^R^ *	*aac(6′)-Iaa*, *aph(6)-Id*, *aph(3″)-Ib*, *bla*_TEM-1B_, *sul2*, *tet(B)*	IncQ1
PT_SE0352	F	1,4,[5],12:i:-	42	366507	None	*aac(6′)-Iaa*, *parC:*T57S	IncI1-I(α)
PT_SE0355	W	Poona	447	366506	STR^I^	*aac(6′)-Iaa*, *parC*:T57S	NF
PT_SE0354	W	Rissen	469	366509	AMP^R^, DOX^R^, STR^R^, SXT^R^, TET^R^ *	*aac(6′)-Iaa*, *aadA1*, *aadA2*, *bla*_TEM-1B_, *drfA12*, *mph(A)*, *parC*:T57S, *qacE*, *sul1*, *tet(A)*	IncQ1
PT_SE0357	W	Saintpaul	50	366508	STR^I^	*aac(6′)-Iaa*	NF

F: fecal; W: water; ^R^ Resistant; ^I^ Intermediate; * MDR profile; AMP: ampicillin; DOX: doxycycline; STR: streptomycin; SXT: sulfamethoxazole–trimethoprim; TET: tetracycline; NF: not found.

## Data Availability

Data are contained within the article and Appendix A.

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
