# Peer review of "Gulls in Porto Coastline as Reservoirs for Salmonella spp.: Findings from 2008 and 2023"

_microorganisms, 2023, doi:10.3390/microorganisms12010059_

Round 1

Reviewer 1 Report

Comments and Suggestions for Authors

The objective of this study was to assess the antimicrobial resistance and molecular profiles of Salmonella spp. isolates obtained from fecal samples of gulls in the city of Porto in 2008 and 2023 and to evaluate environmental contamination, from surface water samples.

The aim and topic (antibiotic resistance) are very interesting, techniques used in this study, primarily WGS, represent newest advanced molecular biology methods.

I have one question about the methodological part.

After selective isolation on HEA and XLD, bacterial isolates from both fecal and water samples underwent biochemical tests to differentiate Salmonella spp. from other non-lactose fermenters of the family Enterobacteriaceae. Why didn't the authors choose MALDI identification instead of biochemical identification at this stage? I would also recommend explaining in more detail to the readers the purpose for which the PCR method was included in the methodology.

In the Results chapter, the last part, Phylogenetic analysis, is incorrectly numbered.

The results of the study are certainly very interesting, but I do not consider them very original.

Author Response

Thank the reviewer for the insightful suggestion. Phenotypic and genotypic tests are complementary and phenotypic antibiotic susceptibility testing remains the gold standard (please refer to the commentary on the topic, 10.2217/fmb-2022-0109). Nonetheless, the authors acknowledge MALDI-TOF MS identification due to its complementarity and high-throughput compared to biochemical testing and will consider its incorporation in future works. Moreover, the authors’ initial choice of using biochemical tests was influenced by our subsequent use of PCR, which played a crucial role in confirming the presence of Salmonella spp. by targeting invA gene. The PCR technique served as a solid foundation before progressing to further advanced molecular techniques (WGS), contributing to the overall robustness of our results. To reflect this, the authors modified the subsection 2.3 title to “Salmonella identification by Polymerase Chain Reaction (PCR)” and added a sentence at the beginning of the second paragraph stating “PCR was chosen over other molecular techniques due to its specificity in Salmonella spp. identification by targeting invA gene”.

The authors also appreciate the reviewer's keen observation regarding the numbering in the Phylogenetic Analysis section. We have corrected the numbering to ensure the proper sequencing and flow of information in the Results chapter.

The authors understand the reviewer’s concern. However, the authors think that our study introduces novel insights in the presence of Salmonella spp. in gull populations, particularly in the city of Porto at two time points separated by 15 years (2008 and 2023), providing a temporal and spatial context that enriches our understanding of Salmonella prevalence in gull populations. The authors also performed a thorough assessment of antimicrobial resistance and molecular profiles, using advanced molecular techniques, such as whole-genome sequencing (WGS). Our research significantly refines the understanding of Salmonella diversity in gull fecal samples and surface water. Therefore, the novelty of this study was added in the introduction (“This study, to the best of our knowledge, is the first to evaluate the presence of Salmonella spp. in gull populations in the city of Porto.”).

Reviewer 2 Report

Comments and Suggestions for Authors

The manuscript deals with antimicrobial resistance and molecular profiles of Salmonella isolates obtained from seagull faeces samples in the city of Porto Portugal, in 2008 and 2023. In addition, surface water samples taken from at the same location in 2023.

Main comments

A number of such studies have been published in the recent past. The presented study does not bring new knowledge to the wider scientific community and has only local significance. Another weakness of this study is its incompleteness. For the study, it would be beneficial to extend it by examining possible local sources of salmonella colonization of seagulls, e.g. sampling wastewater, waste in landfills, etc. The selection of isolates for serotyping, which was based on the antibiogram profile, could have excluded interesting isolates from the study and distorted the overall results.

A significant part of the discussion is devoted to antibiotic policy and the categorization of antimicrobial drugs, which is only partially related to the actual results. The cited data in Figure A1 and A2 could only be used in the case of examined Salmonella isolates from humans and farm animals and the respective wastes.

Minor comments

Table 2: The order of the isolates is not systematic. I recommend arranging alphabetically according to the names of serotypes and within one serotype according to the ST designation.

Figure 5 is not informative when it includes all Salmonella serotypes in a given year.

Author Response

Thank the reviewers for thoughtful review and valuable feedback on our manuscript.

While we understand the potential value of investigating other local sources to assess Salmonella colonization in seagulls, our study was focused on assessing novel insights in the presence of Salmonella spp. in gull populations in two time points separated by 15 years, providing a temporal context that enriches our understanding of its prevalence in gull populations. Therefore, the novelty of this study was added in the introduction (“This study, to the best of our knowledge, is the first to evaluate the presence of Salmonella spp. in gull populations in the city of Porto.”).

Regarding the selection of strains based on antimicrobial profiling, all presumptive Salmonella isolates underwent confirmation by PCR. In each of the 24 sampling instances from both 2008 and 2023, at least one Salmonella isolate per sample was chosen after PCR confirmation. In the year 2008, multiple Salmonella isolates were identified within a single sample, prompting the selection of all isolates exhibiting distinct antibiograms in the same sample. This selection strategy ensured the inclusion of a substantial number of isolates, contributing to the representativeness of the study.

While Figures A1 and A2 exclusively depict isolates from humans and farm animals, we emphasize the sentinel role of seagulls as carriers of Salmonella, which is supported by available literature. Their interactions with the environment expose them to selective pressures from antibiotics used in both human and veterinary medicine. Consequently, the observed resistance in Salmonella from seagulls may act as indicators for the circulation of AMR resulting from antibiotic therapy practices in both human and veterinary medicine.

Responding to the suggestion, the authors have organized the table as recommended, first by the name of the serotype and then by ST.

Regarding Figure 5 (now Figure 2), its objective is to offer a clearer depiction of antimicrobial resistance variations at the specified time points (2008 and 2023) and between fecal and water samples from 2023, although an analysis by serotype could be an interesting secondary approach.

Reviewer 3 Report

Comments and Suggestions for Authors

General: Abstract: It is necessary to improve the information summarizing the presented work. The objective should be clearly described. Additionally, methods used and key results should be summarized. No information is provided on detected genes or plasmids and differences by years of isolation.

Keywords: Keywords need to be reviewed; for example, are Laurus fuscus and Laurus cachinnans wild gulls?

Introduction: A paragraph about the method for Salmonella identification, antibiotic resistance, and its relevance to public health, prevalent resistance and virulence genes, plasmids, and mobile genetic elements should be added.

Materials and Methods: It should be logically organized. For example, section 2.3 should focus on identification rather than the use of PCR. Section 2.5 should be integrated into 2.5.2. The method for detecting virulence factors and plasmids, not included in this section but present in the results, should be added. Section 3.3.4 and figure 2 should be derived from section 2.5.2 and integrated into the cgMLST profiles results.

Results: They should follow a logical order of sampling and identification of isolates, sequence type (ST), and core genome multilocus sequence typing (cgMLST) of WGS data, determination of serotypes, detection of antibiotic resistance and virulence genes, detection of plasmids and mobile genetic elements (MGEs). Tables and figures should be ordered according to a more logical sequence for readers.

Discussion: It needs to be reorganized as it is confusing. The research presented is on Salmonella reservoirs, so L454-464 should be at the beginning. Follow with L44-453, followed by paragraphs from L419-443, ending with L379-418.

Conclusions: Move Figure 5 from here to add it to the AMR results. Connect the results more effectively to their contribution to public health.

Specific: L130-149: Which strains were identified with PCR? Were there dimers in the amplification?

L136-148: Add the amplification product in base pairs generated by the primers as positive. Also, molecular marker size.

L161: Why wasn't Escherichia coli ATCC 25922 used as a control established in CLSI?

L162-163: What type of antibiotics are referred to as MDR?

L219-223: Summarize this information in a table. Use L513 only for the number of samples per year.

L267-276: Add the antigenic formula of monophasic variants in tables 2 and 3.

L336-338: Why was ≤5 used as a cutoff to determine a cluster? Currently, cgMLST typically uses 10 or fewer loci to establish closely related strains. Therefore, mention the reference or established criterion for this decision.

L367: The MST has poor resolution and needs to be redone. Additionally, distances in the number of genes in clusters B1 and B2 are not visible. Add that the figure was made from 3002 target genes as cgMLST.

L467: Add more information about monophasic variants and their relevance in outbreaks in various countries. Also, their relationship with AMR.

L389-390: Since line 169 mentions MDR, were MDR found in the analyzed strains?

L410: Why are MGEs associated with plasmids not included?

L433: This cannot be appreciated in the MST to see the relevance of this and this paragraph.

L511: Supplementary tables and figures should go in a supplementary file, not in this manuscript.

Author Response

General: Abstract: It is necessary to improve the information summarizing the presented work. The objective should be clearly described. Additionally, methods used and key results should be summarized. No information is provided on detected genes or plasmids and differences by years of isolation.

R: Thank the reviewer for the suggestion. The abstract has been revised as follows:

“Gulls act as intermediaries in the exchange of microorganisms between the environment and human settlements, including Salmonella spp. This study assessed antimicrobial resistance and molecular profiles of Salmonella spp. isolates obtained from fecal samples of gulls in the city of Porto, Portugal, in 2008 and 2023, and in water samples in 2023. Antimicrobial susceptibility profiling revealed an improvement in the prevalence (71% to 17%) and antimicrobial resistance between the two collection dates. Two isolate collections from both 2008 and 2023 underwent serotyping and whole genome sequencing, revealing genotypic changes, including an increased frequency of the monophasic variant of S. Typhimurium. qacE was identified 2008 and 2023, for both water and fecal samples, with the most isolates exhibiting a MDR profile. The most frequently observed plasmids type in 2008 were IncF (23%), while IncQ1 predominated in 2023 (43%). Findings suggest that Salmonella spp. circulate between humans, animals, and the environment. However, the genetic heterogeneity among the isolates from gulls’ feces and surface water may indicate a complex ecological and evolutionary dynamic shaped by changing conditions. The observed improvements are likely due to measures to reduce biological contamination and antimicrobial resistance. Nevertheless, additional strategies must be implemented to reduce the public health risk modeled by the dissemination of pathogens by gulls.”.

Keywords: Keywords need to be reviewed; for example, are Laurus fuscus and Laurus cachinnans wild gulls?

R: Thank the reviewer for the suggestion. While both species (Laurus fuscus and Laurus cachinnans) are commonly found in Portugal, and were sampled in our research, we acknowledge that they might not qualify as essential keywords. Therefore, we revisited the keywords and incorporated reviewed the keywords, such as "Laurus spp., whole genome sequencing" and "serotyping" to more accurately represent the methodologies utilized in our study.

Introduction: A paragraph about the method for Salmonella identification, antibiotic resistance, and its relevance to public health, prevalent resistance and virulence genes, plasmids, and mobile genetic elements should be added.

R: The authors appreciate the reviewer’s comment. We think that this introduction is already focused on several topics demanded by the reviewer, namely: Lines 38-47: Salmonella relevance for public health (PH); Lines 50-60: Antimicrobial resistance and the relevance for PH. For the authors, those were the most important topics and we wanted a short and objective introduction. However, we have incorporated additional information In the introduction concerning the identification of Salmonella in Europe, including corresponding antimicrobial resistance data and serotype information (“Indeed, recent studies conducted in northern Europe revealed an occurrence of Salmonella spp. in gulls of approximately 21% and 19.2% of Salmonella isolates exhibited multidrug-resistant (MDR) profiles [22]. Also, Typhimurium was the most frequently observed serotype [22]. In contrast, Italy reported a prevalence of 1.3% of Salmonella spp. in gulls and all strains were identified as Salmonella arizonae, with a predominant resistance to sulphonamides [23].)

Materials and Methods: It should be logically organized. For example, section 2.3 should focus on identification rather than the use of PCR. Section 2.5 should be integrated into 2.5.2. The method for detecting virulence factors and plasmids, not included in this section but present in the results, should be added. Section 3.3.4 and figure 2 should be derived from section 2.5.2 and integrated into the cgMLST profiles results.

R: The authors appreciate the reviewer’s comment. To enhance clarity, section 2.3 was renamed for “Salmonella identification by Polymerase Chain Reaction (PCR)”. The methods for detecting virulence factors and plasmids were already detailed in the section 2.5.2, as follows “Tools from Centre for Genomic and Epidemiology (CGE, http://www.genomicepidemiology.org; accessed on 31 July 2023) were also used to assess antibiotic resistance genes and point mutations (ResFinder 4.1;) and plasmid replicons (PlasmidFinder 2.1). Virulence genes were obtained from the Virulence Factors of Pathogenic Bacteria (VFDB) platform (http://www.mgc.ac.cn/cgi-bin/VFs/genus.cgi?Genus=Salmonella; accessed on 1 July 2023), using the Vfanalyser tool”.

Regarding results chapter, the subsections have been reformulated to be more elucidative for readers. Figure 2 (now Figure 3) is associated with the phylogenetic analysis and has been integrated into a subsection under WGS characterization (3.3.3.). This adjustment aims to create a more seamless and logical flow in the presentation of results.Parte superior do formulário

Results: They should follow a logical order of sampling and identification of isolates, sequence type (ST), and core genome multilocus sequence typing (cgMLST) of WGS data, determination of serotypes, detection of antibiotic resistance and virulence genes, detection of plasmids and mobile genetic elements (MGEs). Tables and figures should be ordered according to a more logical sequence for readers.

R: The authors appreciate the reviewer’s comment. Our results are organized in the same manner as the methods, beginning with the sampling process and overall antimicrobial profiling, followed by a detailed characterization of the Salmonella collections (serotyping, ST, antimicrobial determinants, virulence factors, and plasmids). Additionally, the table has been organized, first by the name of the serotype and then by ST, as suggested Reviewer 2.

Discussion: It needs to be reorganized as it is confusing. The research presented is on Salmonella reservoirs, so L454-464 should be at the beginning. Follow with L44-453, followed by paragraphs from L419-443, ending with L379-418.

R: The discussion is organized into two main topics: global antimicrobial resistance for all Salmonella isolates and an in-depth study of a selected collection comprising 26 isolates from 2008 and 7 from 2023. In this instance, the first three paragraphs address the occurrence of Salmonella and global antimicrobial resistance observed across all isolates. Subsequent sections delve into the in-depth analysis conducted on the selected isolates, covering antimicrobial determinants, serotyping, plasmids, and phylogeny. The concluding two paragraphs shift the focus to the positioning of seagulls in the city of Porto, emphasizing the profound impact of the environment on these birds. This section sheds light on seagull behavior, highlighting their dual role as both vectors and sources of Salmonella. The intricate interplay between seagulls and the environment is discussed, underscoring the valuable role these birds play as indicators. In order to clarify the discussion, some information was added “Previous research has indicated an increased risk of transmitting pathogens, such as Salmonella spp., Campylobacter spp., and Chlamydia spp., between infected gulls and human populations [33]. … Despite European regulations on antibiotic use, MDR isolates were identified in both years, exhibiting a higher relative percentage in 2023 (23% in 2008 and 44% in 2008 and 2023, respectively) but lower absolute frequency (18/77 against 8/18 in 2023). This suggests an increased likelihood of encountering MDR strains, despite an overall improving trend in antimicrobial resistance. … Regarding both collections of 2008 and 2023, the phenotypic antimicrobial sensitivity results and WGS analysis were in alignment… Additionally, a cluster of the monophasic variant of S. Typhimurium was detected in clinical samples from Porto spanning the years 2001 to 2011 [48]. Despite the isolation of 5 isolates of monophasic variant of S. Typhimurium from both collections, only one exhibited multidrug resistance (MDR). It's worth noting that this serotype often exhibits MDR profiles [49,50].”

Conclusions: Move Figure 5 from here to add it to the AMR results. Connect the results more effectively to their contribution to public health.

R: The authors appreciate the reviewers' suggestion. Figure 5 has been re-designated as Figure 2 and relocated to Section 3.2 within the Results chapter.

Specific: L130-149: Which strains were identified with PCR? Were there dimers in the amplification?

R: PCR was specifically designed to identify Salmonella spp. by targeting the invA gene. All strains showing presumptive Salmonella characteristics in the biochemical tests and yielding positive results in the agglutination test underwent PCR confirmation. During the PCR confirmation process, there were no instances of dimer amplification observed.

L136-148: Add the amplification product in base pairs generated by the primers as positive. Also, molecular marker size.

R: The information regarding product size of PCR fragments and molecular marker size was added in the 2.3 section:

“The presence of 284-bp fragments were subjected to electrophoresis on 1.5% (w/v) agarose gel (Agarose Ultrapure grade, NZYTech) in 1xTBE at 100V for 45 min and stained with Green Safe Premium (NZYTech).  Ladder VII (Nzytech) was used as the molecular weight marker.”

L161: Why wasn't Escherichia coli ATCC 25922 used as a control established in CLSI?

R: We apologize for this misunderstanding. For antimicrobial susceptibility testing, we utilized Escherichia coli ATCC 25922 as the control, while Salmonella typhimurium CECT 443 served as the control for the PCR technique. We apologize for any confusion caused by this oversight. Nonetheless, it's worth noting that our laboratory routinely performs quality controls as part of its clinical microbiological work in analyzing veterinary samples for the Veterinary Hospital of University of Porto.

L162-163: What type of antibiotics are referred to as MDR?

R: Bacteria were classified as MDR, when presented resistance to 3 or more antibiotics classes, according to Magiorakos, et al, 2011. In our study, the antibiotics classes studied were: tetracyclines, aminoglycosides, cephalosporins, carbapenems, fluoroquinolones, folate pathway inhibitors, monobactams, penicillins, penicillins and β-lactams inhibitors, phenicols, phosphonic acids and polymyxins.

L219-223: Summarize this information in a table. Use L513 only for the number of samples per year.

R: The information of the section 3.1 was summarized in a table (Table 1) and the text reformulated, as follows “In 2008, a total of seventy-seven isolates were obtained from 17 (71%) fecal samples. While, in 2023, 11 isolates were identified from 4 (17%) fecal samples and 7 recovered from 3 (13%) water samples (Table 1).”.

L267-276: Add the antigenic formula of monophasic variants in tables 2 and 3.

R: The authors are grateful for this suggestion and have added the antigenic formula to the tables.

L336-338: Why was ≤5 used as a cutoff to determine a cluster? Currently, cgMLST typically uses 10 or fewer loci to establish closely related strains. Therefore, mention the reference or established criterion for this decision.

R: The authors based the analysis on the methods implemented in National Reference Laboratories for Salmonella, that recommend the use of 5 AD for the definition of cluster. We appreciate the reviewer’s suggestion and we will include a recent outbreak report as reference (https://www.ecdc.europa.eu/sites/default/files/documents/ROA_S-Enteritidis-ST11_chicken-meat_2023_amended.pdf). 

L367: The MST has poor resolution and needs to be redone. Additionally, distances in the number of genes in clusters B1 and B2 are not visible. Add that the figure was made from 3002 target genes as cgMLST.

R: The image was redone and we have included the original file separately to allow the journal to use the higher resolution file for publication. The information regarding the 3002 loci was added to the figure 3 “The core genome minimum spanning tree was created within the Enterobase pipeline using the NINJA NJ algorithm and GrapeTree tool, comprising a total of 3002 target loci of Salmonella.”

L467: Add more information about monophasic variants and their relevance in outbreaks in various countries. Also, their relationship with AMR.

R: The authors appreciate the reviewers' suggestions, and additional information regarding monophasic variants has been incorporated into the discussion. Specifically, we've highlighted that the monophasic variant of S. Typhimurium often demonstrates multidrug resistance (MDR) profiles, as it follows “Despite the isolation of 5 isolates of monophasic variant of S. Typhimurium from both collections, only one exhibited multidrug resistance (MDR). It's worth noting that this serotype often exhibits MDR profiles.”. Furthermore, the reference linking monophasic variants to outbreaks was reiterated in the context of ST34, stating "In 2023, ST34 was the most frequent type, linked to the monophasic variant of S. Typhimurium [45], which has been associated with foodborne outbreaks in Europe and China [46,47].". Nevertheless, a new sentence was also added to the conclusion, as it follows “An increased frequency of the monophasic variant of S. Typhimurium was also observed, raising public health concerns due to its association with multidrug-resistant (MDR) profiles and outbreaks throughout Europe and China”.

L389-390: Since line 169 mentions MDR, were MDR found in the analyzed strains?

R: MDR analysis were performed. Therefore, a sentence was added in the section 3.2 regarding antimicrobial results (“MDR profiles were observed in 18 isolates (23%) from 2008. In 2023, MDR were detected in 8 (44%) from 2023, of which 6 (55%) were from fecal samples and 2 (29%) from water.”), as well in the discussion (“Despite European regulations on antibiotic use, MDR isolates were identified in both years, exhibiting a higher relative percentage in 2023 (23% in 2008 and 44% in 2023). This suggests an increased likelihood of encountering MDR strains, despite an overall improving trend in antimicrobial resistance.”).

L410: Why are MGEs associated with plasmids not included?

R: The exclusion of Mobile Genetic Elements (MGEs) associated with plasmids from our study was a decision based on the specific focus of our research. The primary objectives were centered around antimicrobial profiling, serotyping and Sequence Type (ST) determination. While we acknowledge the importance of MGE analysis, it falls outside the scope of the current study's objectives

L433: This cannot be appreciated in the MST to see the relevance of this and this paragraph.

R: Although the phylogenetic relationship between the isolates and the complete database of EnteroBase covering the years 2014 and 2018 was not carried out, we believe that the indication that the Rissen ST469 serovar was isolated in the Azores is relevant, thus indicating a possible spread between the islands and mainland Portugal.

L511: Supplementary tables and figures should go in a supplementary file, not in this manuscript.

R: We appreciate your suggestion and we agree with the need to organize supplementary tables and figures in a separate supplementary file. Accordingly, a new file containing all supplementary materials has been created to enhance the clarity and organization of the manuscript.

Reviewer 4 Report

Comments and Suggestions for Authors

The study provides valuable insights into the role of gulls as intermediaries in the exchange of microorganisms, specifically Salmonella spp., between the environment and human habitats. While the comparative analysis of data from 2008 and 2023 offers a compelling overview of the changes in prevalence and antimicrobial resistance, the paper would benefit significantly from a more detailed description of the methodologies employed. Additionally, the investigation into the phenotypic and genotypic alterations, such as the rise of the monophasic variant of S. Typhimurium, presents fascinating findings; however, the paper could be strengthened by a deeper exploration of the implications of these changes for public health and the evolutionary dynamics of Salmonella spp. Lastly, the recommendation for additional strategies to mitigate health risks posed by gull-mediated pathogen dissemination, while valid, lacks specificity. Elaborating on potential measures and their effectiveness would greatly enhance the practical application and impact of this research. However, some of the wording and structuring of the manuscript need to be adjusted to make the message and data clearer. If the following comments could be addressed, it would strengthen the study. Also, although I am not a native speaker, I find the text sometimes hard to follow, and if resubmitted, I would recommend proof-reading.

The following points need to be addressed prior to publication as indicated below.

1. Line 88, A total of 72 samples were analyzed: 24 fecal samples from 2008, 24 fecal samplesfrom 2023, and 24 water samples from 2023. 

A suggestion for improvement would be to consider increasing the number of samples. This increase could enhance the statistical power of your findings and possibly provide a more robust understanding of the prevalence and antimicrobial resistance patterns of Salmonella spp. Furthermore, it would be beneficial if you could compare your results with previous studies in this field. This comparative analysis could significantly enrich the discussion section of your paper, offering readers a clearer understanding of how your findings align with or diverge from established knowledge in the field.

2. Line 348, In reviewing the provided materials, it has been observed that the clarity of the included diagram is suboptimal. This issue detracts from the overall comprehensibility and professional presentation of your work. It is recommended that you enhance the resolution and overall quality of this image.

Author Response

The study provides valuable insights into the role of gulls as intermediaries in the exchange of microorganisms, specifically Salmonella spp., between the environment and human habitats. While the comparative analysis of data from 2008 and 2023 offers a compelling overview of the changes in prevalence and antimicrobial resistance, the paper would benefit significantly from a more detailed description of the methodologies employed. Additionally, the investigation into the phenotypic and genotypic alterations, such as the rise of the monophasic variant of S. Typhimurium, presents fascinating findings; however, the paper could be strengthened by a deeper exploration of the implications of these changes for public health and the evolutionary dynamics of Salmonella spp. Lastly, the recommendation for additional strategies to mitigate health risks posed by gull-mediated pathogen dissemination, while valid, lacks specificity. Elaborating on potential measures and their effectiveness would greatly enhance the practical application and impact of this research. However, some of the wording and structuring of the manuscript need to be adjusted to make the message and data clearer. If the following comments could be addressed, it would strengthen the study. Also, although I am not a native speaker, I find the text sometimes hard to follow, and if resubmitted, I would recommend proof-reading.

R: Thank you for your thoughtful feedback. The methods chapter were reformulated, especially the subsection 2.3. Salmonella identification by Polymerase Chain Reaction (PCR). While we understand the potential value of investigating other local sources to assess Salmonella colonization in seagulls, our study was focused on introducing novel insights in the presence of Salmonella spp. in gull populations in two time points separated by 15 years, providing a temporal context that enriches our understanding of Salmonella prevalence in gull populations. 

Regarding the elaboration of potential measures and their effectiveness was already addressed in conclusion chapter, as it follows “Limiting the exposure of gulls to biological contamination and implementing population control in areas of high population density, by reducing scavenging opportunities for example, could help reduce the public health risk posed by these reservoirs and/or vectors of pathogenic agents.”

Nevertheless, the manuscript has been revised by a C2 level English speaker and, additionally, using a grammar correction software (Grammarly).

The following points need to be addressed prior to publication as indicated below.

  1. Line 88, “A total of 72 samples were analyzed: 24 fecal samples from 2008, 24 fecal samples from 2023, and 24 water samples from 2023.”

A suggestion for improvement would be to consider increasing the number of samples. This increase could enhance the statistical power of your findings and possibly provide a more robust understanding of the prevalence and antimicrobial resistance patterns of Salmonella spp. Furthermore, it would be beneficial if you could compare your results with previous studies in this field. This comparative analysis could significantly enrich the discussion section of your paper, offering readers a clearer understanding of how your findings align with or diverge from established knowledge in the field.

R: We appreciate your suggestions. Despite the lack of statistical significance, our work has found general trends in AMR. Moreover, increasing the number of samples would add a third sampling time (2024) to the study and not expand on previous samples, which were collected in 2008 and early 2023. Additional sample collections would not improve the statistical power of previous results and add complexity to the Fisher exact test by adding an additional category. Acknowledging the potential advantages of a larger sample size, we will consider collecting a higher number of samples in future research.

Additionally, we recognize the importance of contextualizing our results in the broader scientific literature, however, we believe it is paramount to emphasize the significance of framing the observed resistance in seagulls within the context of European policies regarding antibiotic therapy and its impact on the seagulls, adopting a “One health” perspective. Therefore, we have added two more studies regarding research on Salmonella spp. in Porto, which were referenced in discussion “…Nonetheless, in 2013, Salmonella spp. have been isolated in two lakes of Porto, demonstrating high susceptibility to all antibiotics tested…. Additionally, a cluster of the monophasic variant of S. Typhimurium was detected in clinical samples from Porto spanning the years 2001 to 2011.”

  1. Line 348, In reviewing the provided materials, it has been observed that the clarity of the included diagram is suboptimal. This issue detracts from the overall comprehensibility and professional presentation of your work. It is recommended that you enhance the resolution and overall quality of this image.

R: Thank you your suggestion. The Figure 3 was reuploaded in order to improve overall quality. We have included the original file separately to allow the journal to use the higher resolution file for publication.

Round 2

Reviewer 3 Report

Comments and Suggestions for Authors

Line 77 is repeated with line 78 in the new version